:ᗢ: PLOS | ONE

**Data Availability Statement:** All relevant data are within the paper and its Supporting Information files.

# m$^6$A minimally impacts the structure, dynamics, and Rev ARM binding properties of HIV-1 RRE stem IIB

Chia-Chieh Chu[ORCID][1], Bei Liu[1], Raphael Plangger[2], Christoph Kreutz[2], Hashim M. Al-Hashimi[1,3]*

1 Department of Biochemistry, Duke University School of Medicine, Durham, NC, United States of America, 2 Institute of Organic Chemistry and Center for Molecular Biosciences CMBI, Universität Innsbruck, Innsbruck, Austria, 3 Department of Chemistry, Duke University, Durham, NC, United States of America

* hashim.al.hashimi@duke.edu

## Abstract

$N^6$-methyladenosine (m$^6$A) is a ubiquitous RNA post-transcriptional modification found in coding as well as non-coding RNAs. m$^6$A has also been found in viral RNAs where it is proposed to modulate host-pathogen interactions. Two m$^6$A sites have been reported in the HIV-1 Rev response element (RRE) stem IIB, one of which was shown to enhance binding to the viral protein Rev and viral RNA export. However, because these m$^6$A sites have not been observed in other studies mapping m$^6$A in HIV-1 RNA, their significance remains to be firmly established. Here, using optical melting experiments, NMR spectroscopy, and *in vitro* binding assays, we show that m$^6$A minimally impacts the stability, structure, and dynamics of RRE stem IIB as well as its binding affinity to the Rev arginine-rich-motif (ARM) *in vitro*. Our results indicate that if present in stem IIB, m$^6$A is unlikely to substantially alter the conformational properties of the RNA. Our results add to a growing view that the impact of m$^6$A on RNA depends on sequence context and Mg$^{2+}$.

## Introduction

$N^6$-methyladenosine (m$^6$A) is an abundant reversible epitranscriptomic modification found in coding and non-coding RNAs [1–4]. It plays important roles in RNA metabolism [5–8] and is implicated in a growing number of cellular processes [9–15]. m$^6$A has also been found in viral RNAs where it is proposed to modulate host-pathogen interactions [16–19].

A number of studies have reported m$^6$A in the HIV-1 RNA genome [18, 20, 21]. One study examining HIV-1 infected human T cells [20] reported two m$^6$A sites (A68 and A62, Fig 1) in the Rev response element (RRE) stem IIB. RRE is a ~350 nt cis-acting RNA element that is recognized by viral Rev protein to promote the export of unspliced or partially spliced viral RNA to express the structural proteins required for viral replication [22–24]. The two m$^6$A sites were found in stem IIB (RREIIB), which is the primary binding site for Rev [23, 25–27]. Knocking down the methyltransferase complex (METTL3/METTL14) was shown to suppress Rev-RRE mediated RNA export and viral replication, and point substitution mutation of one

**Funding:** This work was supported by the US National Institutes of Health [P50 GM103297 to H. M.A.]; Austrian Science Fund [P28725 and P30370 to C.K.] and Austrian Research Promotion Agency FFG (West Austrian BioNMR, 858017 to C.K.). Funding for open access charge: US National Institutes of Health [P50 GM103297]. The funders had no role in study design, data collection and analysis, decision to publish, or preparation of the manuscript.

**Competing interests:** H.M.A. is an advisor and holds an ownership interest in Nymirum Inc., an RNA-based drug discovery company. This does not alter our adherence to PLOS ONE policies on sharing data and materials.

of the two highly conserved adenines (A68) strongly suppressed viral replication ($> 90\%$) [20]. It was proposed [20] that the methyl group of m6A68 may interact with Rev protein to stabilize Rev-RRE binding, and/or that m6A68 may alter the conformational properties of stem IIB to facilitate Rev recognition. Another different study employing distinct cell lines and mapping methods did not observe these m6A sites on RREIIB suggesting that m6A can enhance HIV-1 replication and mRNA expression through recruitment of the m6A reader proteins YTH-domain containing family (YTHDF) [21]. A third study found m6A in RRE but did not identify the specific site [18]. The study proposed that the YTHDF proteins inhibit HIV-1 replication and infection by blocking viral reverse transcription.

Here, we asked whether methylation of RREIIB with m6A leads to changes in its conformational and Rev arginine rich motif (ARM) binding properties. We were driven to test this hypothesis because our recent studies showed that the internal loop region of RREIIB near the A68 bulge is highly flexible, and can adopt conformations with alternative secondary structures (Fig 1A) that have different export activities [28, 29]. By redistributing this dynamic ensemble of RREIIB, m6A could potentially impact the Rev-RRE interaction and RNA export. Prior studies have shown that m6A can reshape RNA-protein and RNA-RNA interactions by modulating RNA structure [30]. For example, a single m6A was shown to destabilize RNA duplexes by 0.5–1.7 kcal/mol [31, 32] thus enhancing the binding affinity of proteins to their single-stranded RNA targets [33]. The modification destabilizes A-U base pairs because hydrogen bonding requires that the $N^6$-methyl group adopt the unfavorable *anti* conformation [34, 35]. m6A has also been shown to disrupt the non-canonical sheared G-A base pairs to block the assembly of the box C/D snoRNP complexes [36].

We find that modification of the A68 bulge has little effect on the stability, structure, dynamics, as well as the Rev-ARM binding properties of RREIIB. The results indicate that if RREIIB is m6A modified, it is unlikely to substantially change the conformational properties of RRE although we cannot rule out that small changes in the conformational properties could modulate Rev-RRE binding *in vivo*. The results also add to a growing view that the impact of m6A on RNA structure depends on sequence context and $Mg^{2+}$ [31, 32, 36, 37]. For example, while m6A has been shown to destabilize canonical duplexes [31, 32], it can stabilize junctional A-U base pairs in a $Mg^{2+}$ and secondary structure dependent manner [37] as well as in contexts in which the m6A is in a dangling end [32].

## Materials and methods

### Preparation of RNA samples for NMR studies

RREIIB and m6A modified RREIIB (RREIIB^m6A68 and RREIIB^m6A26m6A68) were chemically synthesized using an in-house oligo synthesizer (MerMade 6, BioAutomation) with solid-phase RNA synthesis using N-acetyl protected 2'-tBDSilyl-phosphoramidites (ChemGenes Corporation) and 1 μmol standard columns (1000 Å, BioAutomation) with 4,4'-dimethoxytrityl (DMT)-off synthesis followed by base and 2'-O deprotection (100 μmol DMSO, 125 μL TEA•3HF, heat at 65ºC for 2.5hrs), and ethanol precipitation[28]. A similar approach was used to synthesize site-labeled 15N3-U72-RREII^m6A68 three-way junction using 15N3-uridine phosphoramidites [38] and using DMT-off 2'-O deprotection to obtain cleaner NMR spectra.

All RNA samples were purified using 20% (w/v) denaturing polyacrylamide (29:1) gel within 8M urea, 20 mM Tris Borate and 1 mM ethylene-diaminetetraacetate (EDTA) TBE buffer followed by Elutrap electro-elution system (Whatmann, GE healthcare) with 40 mM Tris Acetate and 1 mM EDTA (TAE) buffer then ethanol precipitation. The RNA pellets were dissolved in water and annealed by heating at 95°C for 10 mins then rapidly cooling on ice. After measuring the concentration, the RNA samples were buffer-exchanged into NMR buffer

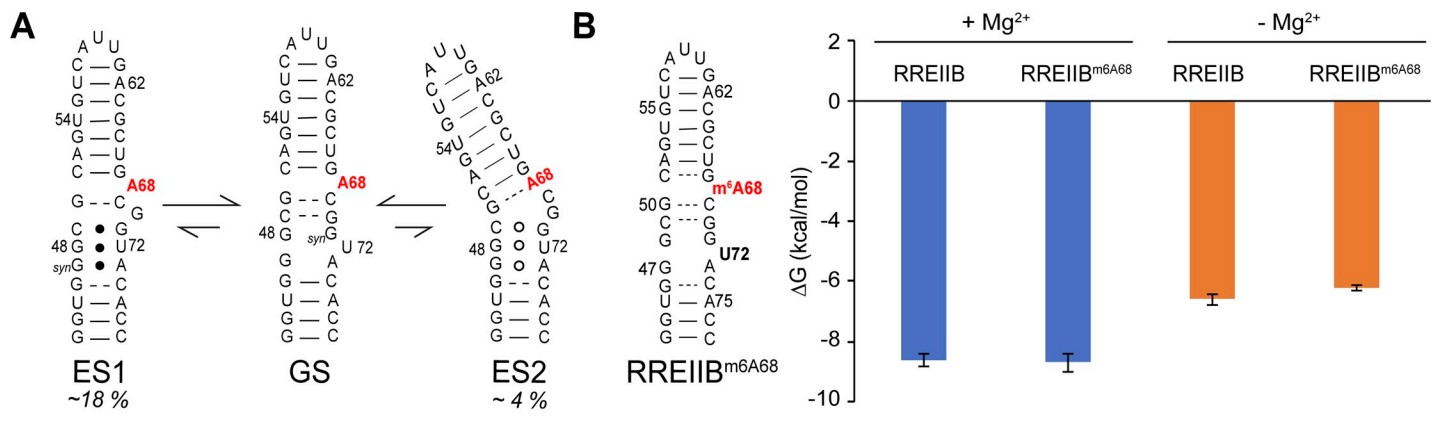

**Fig 1. m⁶A has limited impact on the thermal stability of RREIIB stem loop.** (A) The secondary structure with the proposed N⁶-methyl modified A68 bulge (highlighted in red) of RREIIB native GS and two non-native ES1 and ES2. (B) Impact of m⁶A68 on the thermal stability of the RREIIB hairpins with and without 3 mM Mg²⁺. Shown are the free energy of annealing of the methylated and unmodified RREIIB with and without Mg²⁺. (C) RNA thermodynamic parameters from UV melting. Data are presented as the mean ± standard deviation of at least three independent measurements. Uncertainty in the calculated thermodynamic parameters were determined by error propagation as previously described [48].

(15 mM sodium phosphate, 25 mM NaCl, 0.1 mM EDTA, with or without 3 mM MgCl₂ at pH = 6.4) three times using 3kDa Amicon Ultra centrifugal filters (EMD Millipore). The final RNA concentrations were ~1.3 mM and ~0.4 mM for RREIIB and RREII, respectively.

## NMR experiments

NMR experiments were performed on Bruker Avance III 600 MHz or 700 MHz NMR spectrometers equipped with 5 mm triple-resonance cryogenic probes at 25°C. 2D [¹H,¹³C] (C2/C6/C8-H2/H6/H8) and 2D [¹H,¹⁵N] (N1/N3-H1/H3) heteronuclear single quantum coherence (HSQC) spectra were recorded in the absence or presence of 3 mM Mg²⁺ at 25 °C in NMR buffer with 10% D₂O. All NMR data were analyzed using NMRPipe [39] and SPARKY (T.D. Goddard and D.G. Kneller, SPARKY 3, University of California, San Francisco). Unless stated otherwise, all NMR spectra were collected in the presence of Mg²⁺. Resonance intensities were measured using 2D [¹H–¹³C] HSQC experiments. The intensity for each type of C–H spin was normalized to a value of 0.1 using A52-C8H8, A52-C2H2 and U56-C6H6, respectively.

## UV melting experiments

Thermal melting experiments were performed on m⁶A modified and unmodified RREIIB using a PerkinElmer Lambda 25 UV/VIS spectrometer equipped with an RTP 6 Peltier Temperature Programmer and a PCB 1500 Water Peltier System. All RNA samples were buffer exchanged into NMR buffer. RNA samples (concentration ~ 1 mM) were then diluted (with NMR buffer) to 3 μM prior to UV melting measurement, which were performed in triplicate (or more) using a sample volume of 400 μL in a Teflon-stopped 1 cm path length quartz cell.

The absorbance at 260 nm was monitored while the temperature was varied between 15 and 95˚C at a rate of 1C/min. Thermodynamic parameters were obtained by fitting the UV melting curves using nonlinear model fitting in Mathematica 10.0 (Wolfram Research) as previously described [37].

## Fluorescence polarization binding assays

Fluorescence polarization (A) was measured using a CLARIOstar plate reader (BMG LAB-TECH) using 480 nm excitation and a 540 nm emission filter [28, 40]. Fluorescence polarization binding assays were carried out using 3'-end fluorescein labeled Rev-ARM peptide (Rev-Fl, TRQARRNRRRRWRERQRAAAACK-FITC, LifeTein LLC). The serially diluted RNA in the reaction buffer (30 mM HEPES, pH = 7.0, 100 mM KCl, 10mM sodium phosphate, 10 mM ammonium acetate, 10 mM guanidinium chloride, 2 mM MgCl$_2$, 20 mM NaCl, 0.5 mM EDTA, and 0.001% (v/v) Triton-X100) was incrementally added into a 384-well plate containing 10 nM Rev-Fl [40]. The binding curves were fitted to single-site binding equation using least-squares methods implemented in Mathematica 10.0 (Wolfram Research).

$$A = A_{free} + \left(A_{bround} - A_{free}\right) \times \left[\frac{R_T + L_T + K_d - \sqrt{\left(R_T + L_T + K_d\right)^2 - 4R_T \times L_T}}{2R_T}\right]$$

where A is the measured fluorescence polarization; A$_{free}$ is the polarization without Rev-Fl binding; A$_{bound}$ is the polarization with saturated Rev-Fl binding; R$_T$ is the total RNA concentration; L$_T$ is the total Rev-Fl concentration; K$_d$ is the dissociation constant. The uncertainty in (A) was deduced based on the standard deviation over triplicate measurements.

## Results

### m⁶A68 does not alter the thermal stability of RREIIB

We first used optical melting experiments to examine whether methylation of A68 impacts the thermal stability of RREIIB. All experiments were performed in the presence of 3 mM Mg$^{2+}$ unless stated otherwise. This is important given the impact of Mg$^{2+}$ on RNA folding and dynamics [41] and also given recent studies showing that the impact of m⁶A on RNA structural dynamics and stability can depend on Mg$^{2+}$ [37]. For these experiments, we used a stem IIB construct (Fig 1B) containing the wild-type sequence that was recently shown to recapitulate the conformation of stem IIB in the larger three-way junction context [28]. Prior X-ray [42, 43], NMR [44, 45], SAXS [46] as well as chemical probing data on larger fragments of the HIV genome (~350 nt) [47] indicate that RREII stem IIB adopts the predominant secondary structure shown in Fig 1A. In the dominant RREIIB ground state (GS) conformation, A68 adopts an unpaired bulged conformation. When located in bulged nucleotides, m⁶A has previously been shown to slightly destabilize RNA hairpins by 0.4–0.7 kcal/mol [37] most likely due to the disruption of stacking interactions in the flipped-in conformation. Based on the UV melting data, m⁶A68 had a negligible effect on RREIIB stability (Fig 1B and 1C). Interestingly, a small degree of destabilization (by ~0.4 kcal/mol) was observed in the absence of Mg$^{2+}$, most likely because A68 adopts a partially flipped in conformation in the absence of Mg$^{2+}$, and such a conformation could be more susceptible to destabilization by m⁶A [28]. Thus, the m⁶A effect is overridden by Mg$^{2+}$. Indeed, Mg$^{2+}$ stabilizes RREIIB by ~2 kcal/mol while m⁶A only destabilizes it by ~0.4 kcal/mol (Fig 1C).

## m⁶A68 minimally affects structural and dynamic properties of the RREIIB ground state

We used NMR to examine whether methylation of A68 affects the conformation of RREIIB. A single resonance was observed for the N⁶-methyl group in 1D ¹H NMR spectra of RREIIB$^{m6A68}$, consistent with a single dominant conformation for m⁶A68 (Fig 2A). The 1D ¹H imino spectrum

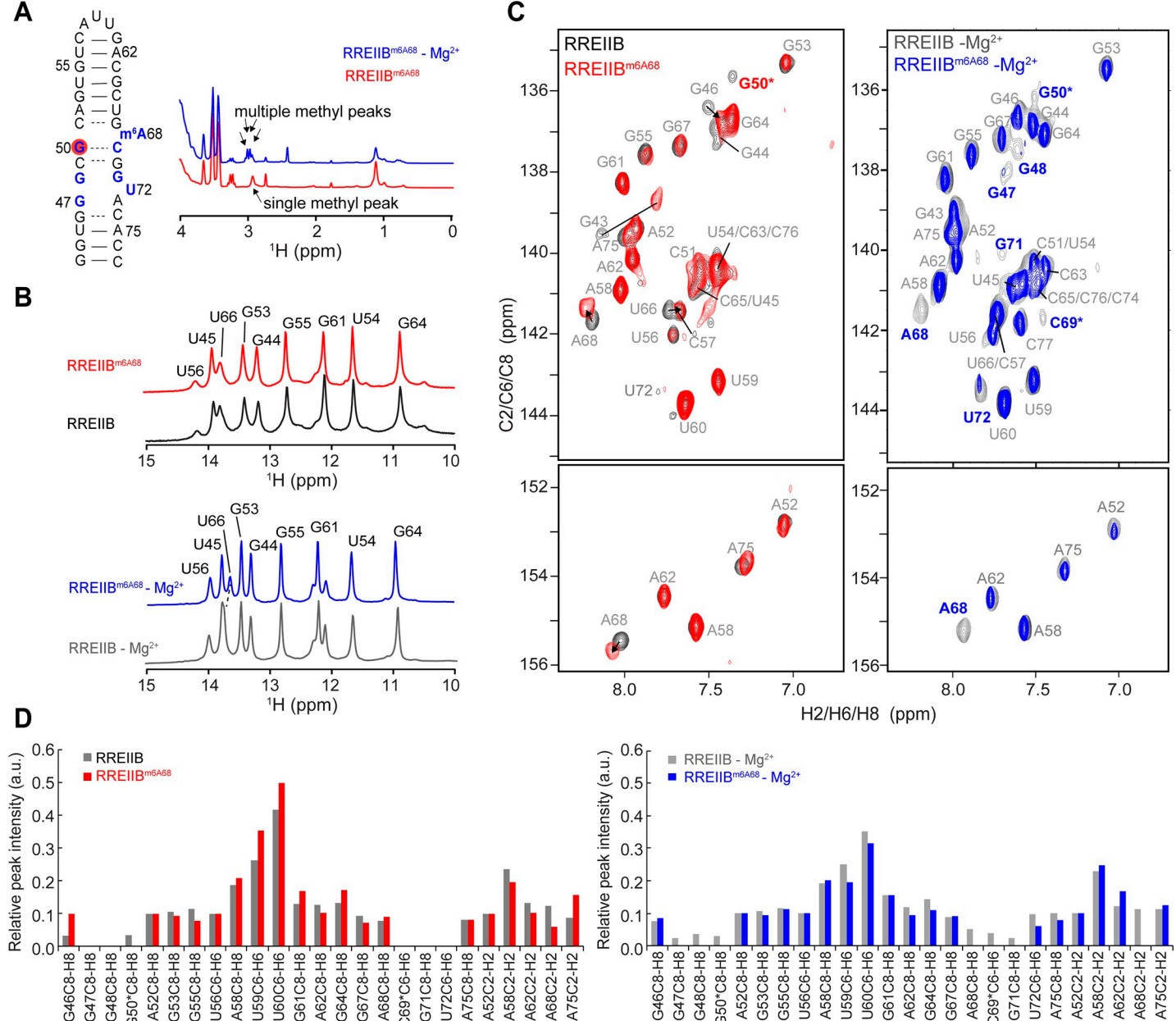

**Fig 2. The m⁶A modification has minor effects on RREIIB structure and dynamics.** (A) Secondary structure of RRE2Bm$^{6A68}$, with the residues showing line-broadening with m⁶A, highlighted in red (with Mg$^{2+}$) and blue (no Mg$^{2+}$). Comparison of the 1D ¹H NMR spectrum of RREIIB$^{m6A68}$ with or without Mg$^{2+}$ with the methyl peak indicated by arrows. The comparison of 1D imino spectra (B) and 2D [¹H,¹³C]-HSQC spectra (C) of RREIIB$^{m6A68}$ and RREIIB in the presence (red) and absence (blue) of 3 mM Mg$^{2+}$. Resonances exhibiting shifting are indicated using arrows, and those with ambiguous assignments denoted using an asterisk. (D) Normalized resonance intensities in 2D [¹H,¹³C]-HSQC spectra of RREIIB$^{m6A68}$ and RREIIB in the presence (red) and absence (blue) of 3 mM Mg$^{2+}$. A52-C8H8, A52-C2H2 and U56-C6H6 were used as a reference and normalized to 0.1. The sample conditions were 1.2–1.5 mM RREIIB$^{m6A68}$ or RREIIB in 15 mM sodium phosphate, 25 mM NaCl, 0.1 mM EDTA, pH 6.4 with or without 3 mM MgCl₂.

of RREIIB$^{m6A68}$ is virtually identical to its unmodified counterpart, indicating that the methylation does not alter the RREIIB secondary structure (Fig 2B). Very good agreement was also observed between the 2D [$^{13}$C,$^1$H] HSQC spectra of modified and unmodified RREIIB with only few residues in the internal loop region (A68, U66, G50, G46) showing minor chemical shift perturbations (Fig 2C). These results indicate that m$^6$A68 modification does not substantially affect the structure of RREIIB GS.

Prior studies showed that m$^6$A can impact RNA conformation in a Mg$^{2+}$ dependent manner [37]. Interestingly, the modification had a larger impact on NMR spectra of RREIIB recorded in the absence of Mg$^{2+}$. Multiple resonances are observed for the N$^6$-methyl group, indicating the co-existence of multiple conformations (Fig 2A). The methylation also induced larger perturbations at A52-U66 base pair near the m$^6$A68 bulge (Fig 2B). In 2D [$^1$H, $^{13}$C] HSQC spectra, the modification induced severe line broadening [49] for resonances belonging to residues G47, G48, G50, A68, C69, G71 and U72 in the internal loop, consistent with enhanced dynamics at the micro-to-millisecond timescales, where only G50 resonance shows broadening in the presence of Mg$^{2+}$ (Fig 2C and 2D). It is likely that the m$^6$A induced line broadening in the absence of Mg$^{2+}$ is due to a changes in ES kinetics or populations. The UV melting data showing that m$^6$A68 destabilizes RREIIB GS in the absence of Mg$^{2+}$ suggests that the enhanced ES dynamics could arise from destabilization of the GS in the absence of Mg$^{2+}$. Similar NMR results were obtained for a double m$^6$A modified RREIIB (RREIIB$^{m6A62m6A68}$) (S1 Fig).

## m$^6$A68 minimally affects the structural and dynamic properties of the excited states in the more native RREII three-way junction

We recently showed that RREII transiently adopts two low-abundance alternative secondary structures ('excited states', ES) referred to as ES1 and ES2 (Fig 3A). A68 remains as a bulge in ES1 while forms a G50-A68 mismatch in ES2. The combined m$^6$A effects to A68 bulge in ES1 and to the specific G(*anti*)-A(*anti*) mismatch in ES2 remain unknown. We therefore examined whether methylation of A68 impacts the dynamics between the GS and ESs in more native RREII three-way junction. This conformational exchange can in principle be measured quantitatively with the use of NMR $R_{1\rho}$ relaxation dispersion (RD) data [28, 50–52]. However, in practice, severe line-broadening and resonance overlap in NMR spectra of the more native three-way junction (RREII) in the presence of Mg$^{2+}$ complicates measurements [28]. We therefore turned to an alternative approach which we recently developed which uses site-specific stable isotope labeling to directly observe the low-populated ES that form under slow exchange kinetics [28]. With this approach, we were previously able to directly observe imino resonances belonging to ES1 and ES2 in RREII, which slows down the exchange kinetics relative to RREIIB (Fig 3A) [28]. In particular, by site-specifically labeling $^{15}$N3-U72, we observed the G48-U72 mismatch, which uniquely forms in both ES1 and ES2, based on the characteristic imino chemical shift of G-U mismatches (Fig 3A) [28].

We used the above strategy to examine how methylation of A68 impacts the GS-ES exchange in RREII. Samples of m$^6$A68 modified and unmodified RREII were chemically synthesized with site-labeled $^{15}$N3-U72 ($^{15}$N3-U72-RREII$^{m6A68}$ and $^{15}$N3-U72-RREII, respectively). The 1D $^1$H imino spectra of modified and unmodified RREII were very similar (Fig 3B) indicating that the modification minimally impacts the GS secondary structure even in the more native three-way junction context. The 2D [$^1$H,$^{15}$N] HSQC spectrum of RREII$^{m6A68}$ includes a resonance characteristic of the ES G48-U72 mismatch, and it shows excellent overlap with the corresponding resonance observed in unmodified RREII (Fig 3C). No other resonances were observed indicating that m$^6$A does not lead to stabilization of alternative

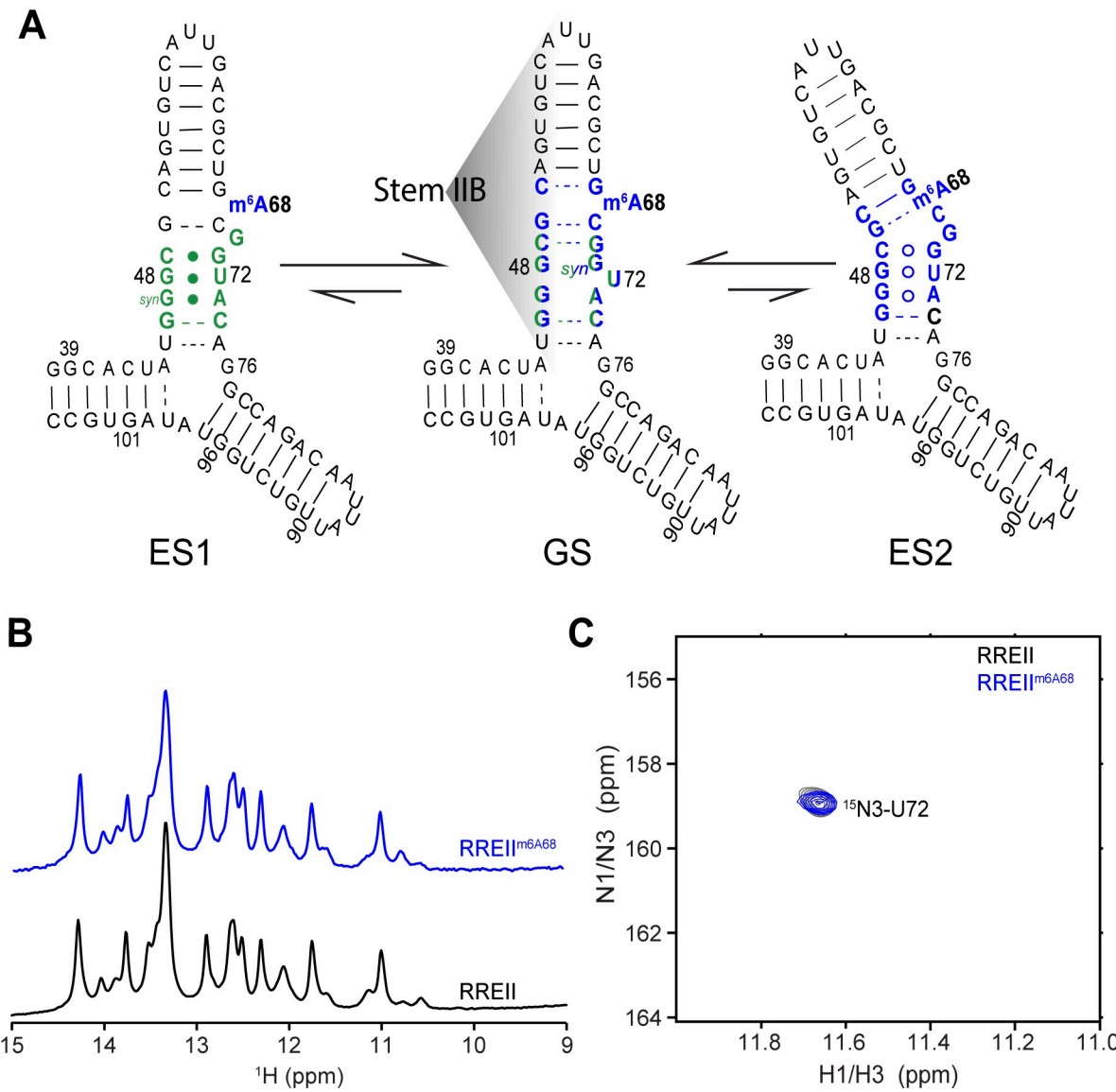

**Fig 3. Selective site-labeling strategy to probe conformational exchange in larger RREII^m6A68 three-way junction.** (A) Proposed secondary structure for ES1 and ES2 in RREII based on ESs observed in RREIIB. Nucleotides that experience exchange due to ES1 and ES2 are colored green and blue, respectively. (B) The comparison of 1D imino spectrum of RREII^m6A68 and RREII. (C) 2D [¹H,¹⁵N]-HSQC spectra of site-specifically labeled ¹⁵N3-U72-RREII^m6A68 and ¹⁵N3-U72-RREII at 25°C showing a single imino resonance at the characteristic chemical shift region (~11.6 ppm) expected for a G-U in ESs. The sample conditions were 0.3–0.5 mM RNA in 15 mM sodium phosphate, 25 mM NaCl, 0.1 mM EDTA at pH 6.4 with 3 mM MgCl₂.

conformations in which U72 is base paired. Similar results were obtained in the absence of Mg²⁺ (S2 Fig). These results indicate that m⁶A68 does not significantly impact structural properties of the GS or the populations of the ESs in the native three-way junction both in the presence and absence of Mg²⁺. Note that we cannot entirely rule out that m⁶A destabilizes ES2 or ES1 and that the observed G48-U72 mismatch reflects either ES1 or ES2, respectively.

## m⁶A68 has a negligible effect on Rev-RRM binding to RREIIB

We used a fluorescence polarization binding assay to examine whether methylation of A68 impacts binding of fluorescein labeled Rev-ARM peptide (Rev-Fl) to RREIIB (Fig 4) [28, 40].

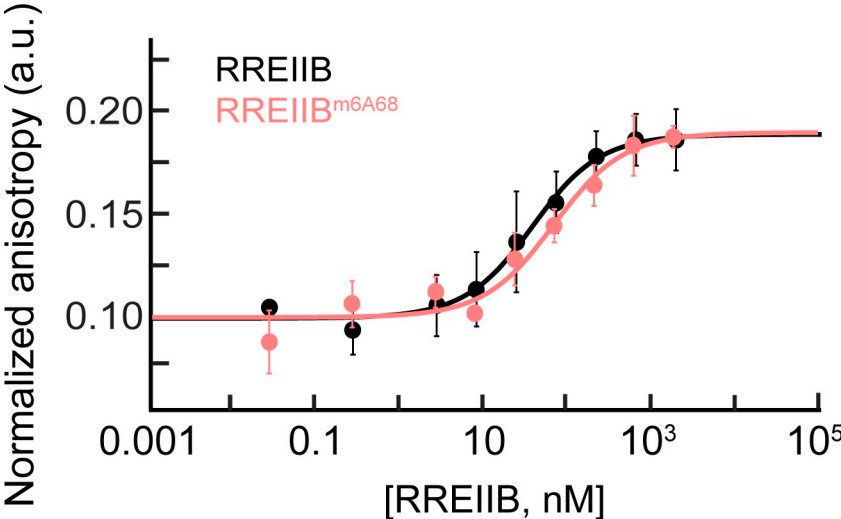

**Fig 4. Measurement of the binding affinity of Rev-ARM peptide to m$^6$A modified or unmodified RREIIB using fluorescence polarization.** Normalized anisotropy values measured for RREIIB$^{m6A68}$ and RREIIB fitted with one-site binding model (see 'Materials and Methods' section). The anisotropy value observed in the absence of RNA was normalized to 0.1. Uncertainty reflects the standard deviation from three independent measurements. Buffer conditions: 30 mM HEPES, pH = 7.0, 100 mM KCl, 10mM sodium phosphate, 10 mM ammonium acetate, 10 mM guanidinium chloride, 2 mM MgCl$_2$, 20 mM NaCl, 0.5 mM EDTA and 0.001% (v/v) Triton-X100. Concentration of Rev-Fl peptide was 10 nM.

Unmodified RREIIB binds to Rev-ARM peptide with apparent K$_d$ = 30.6 ± 5.8 nM, in agreement with prior studies [44, 53]. The binding affinity decreased two-fold (K$_d$ = 62.2 ± 23.8 nM) for the modified RREIIB$^{m6A68}$, indicating that the binding affinity for m$^6$A modified was only slightly weakened relative to unmodified RREIIB especially when considering the uncertainty.

## Discussions

Methylation of RREIIB can promote Rev-RRE interaction by changing the RREIIB conformation so as to favorably bind to Rev or by promoting Rev-RRE binding through direct interaction involving the methyl group. Our results argue against a significant impact on RREIIB structure, as might be expected given placement of m$^6$A68 in the bulge. In addition, the modification slightly weakened binding of the Rev-ARM, and this is consistent with prior NMR [54] and X-ray crystallography [55] studies showing that A68 bulge interacts with Rev protein through non-specific Van Der Waals contacts, and that deleting A68 has minor effects on Rev or Rev-ARM binding affinity to RRE stem IIB [26, 56]. Our data cannot rule out that the exposed and accessible m$^6$A enhances binding in the context of full-length Rev and RRE or promotes RNA export through the recruitment of other host export factors, and that this in turn gives rise to the reported ~2–3 fold decrease in HIV viral RNA pull down using Rev when knocking down METTL3/METTL14 responsible for producing m$^6$A [20]. It is also possible that the structural and dynamic properties of m$^6$A modified RREIIB differ *in vivo* relative to *in vitro*. A recent study [29] showed that the key dynamic properties of unmodified RREIIB are similar *in vitro* and in cells. Further studies are needed to address these possibilities.

The modification had a greater effect on the stability and conformation of RREIIB in the absence of Mg$^{2+}$. We previously showed that Mg$^{2+}$ redistributes the RREIIB ensemble by stabilizing the GS and ES1 relative to ES2 [28]. Our results indicate that Mg$^{2+}$ has a larger effect on the relative stability of these different conformations as compared to the methylation. This

underscores the importance of studying the impact of $m^6A$ modifications in the presence of $Mg^{2+}$ as in a more physiological conditions.

## Supporting information

**S1 Fig. The double $m^6A$ modifications have minor effects on RREIIB structure and dynamics in the presence of $Mg^{2+}$.** (A) Secondary structure of RRE2B$^{m6A62,68}$. Resonances exhibiting line-broadening and perturbations in 2D [$^1H$, $^{13}C$] aromatic HSQC spectra are shown in orange (with $Mg^{2+}$) and green (no $Mg^{2+}$), respectively. The comparison of 1D imino spectra (B) and 2D [$^1H$,$^{13}C$]-HSQC spectra (C) of RREIIB$^{m6A62m6A68}$ and RREIIB in the presence (orange) and absence (green) of 3 mM $Mg^{2+}$. Arrows indicate chemical shift perturbations while ambiguous assignments are denoted using an asterisk.
(TIF)

**S2 Fig. Selective site-labeling strategy to probe conformational exchange in larger 3-way junction RREII$^{m6A68}$ in the absence of $Mg^{2+}$.** (A) The comparison of 1D imino spectrum of RREII$^{m6A68}$ and RREII without $Mg^{2+}$. (B) 2D [$^1H$,$^{15}N$]-HSQC spectra of site-specifically labeled $^{15}N3$-U72-RREII$^{m6A68}$ and $^{15}N3$-U72-RREII without $Mg^{2+}$ at 25˚C showing a single imino resonance at the characteristic chemical shift region (~11.6 ppm) expected for a G-U bp in ES1 and ES2. The sample conditions were 0.3–0.5 mM RNA in 15 mM sodium phosphate, 25 mM NaCl, 0.1 mM EDTA at pH 6.4.
(TIF)

## Acknowledgments

We thank Atul Kaushik Rangadurai for critical comments and Dr. Richard Brennan in Duke University for providing the UV-Vis spectrophotometer. We acknowledge the Duke Magnetic Resonance Spectroscopy Center for supporting NMR experiments, and thank Duke Center of RNA Biology for supporting fluorescence polarization measurements.

## Author Contributions

**Conceptualization:** Chia-Chieh Chu, Hashim M. Al-Hashimi.

**Formal analysis:** Chia-Chieh Chu, Bei Liu, Hashim M. Al-Hashimi.

**Funding acquisition:** Christoph Kreutz, Hashim M. Al-Hashimi.

**Investigation:** Chia-Chieh Chu, Bei Liu, Raphael Plangger, Christoph Kreutz, Hashim M. Al-Hashimi.

**Methodology:** Chia-Chieh Chu, Bei Liu, Raphael Plangger, Christoph Kreutz, Hashim M. Al-Hashimi.

**Project administration:** Hashim M. Al-Hashimi.

**Resources:** Christoph Kreutz, Hashim M. Al-Hashimi.

**Supervision:** Christoph Kreutz, Hashim M. Al-Hashimi.

**Validation:** Chia-Chieh Chu, Bei Liu, Raphael Plangger, Christoph Kreutz, Hashim M. Al-Hashimi.

**Visualization:** Chia-Chieh Chu, Hashim M. Al-Hashimi.

**Writing – original draft:** Chia-Chieh Chu, Hashim M. Al-Hashimi.

**Writing – review & editing:** Chia-Chieh Chu, Hashim M. Al-Hashimi.

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
