## [Decision Letter · Decision Letter 0]

5 Nov 2019

PONE-D-19-28951

m6A minimally impacts the structure, dynamics, and Rev ARM binding properties of HIV-1 RRE stem IIB

PLOS ONE

Dear Dr. Hashim M. Al-Hashimi,

Thank you for submitting your manuscript to PLOS ONE. After careful consideration, we feel that it has great merit but does not fully meet PLOS ONE’s publication criteria as it currently stands. Therefore, we invite you to submit a revised version of the manuscript that addresses the points raised during the review process. Your manuscript has been revised by four reviewers with two of them indicating minor changes indicated below.

We would appreciate receiving your revised manuscript by December 06, 2019. To enhance the reproducibility of your results, we recommend that if applicable you deposit your laboratory protocols in protocols.io, where a protocol can be assigned its own identifier (DOI) such that it can be cited independently in the future. For instructions see: http://journals.plos.org/plosone/s/submission-guidelines#loc-laboratory-protocols

We look forward to receiving your revised manuscript.

Kind regards,

Joanna Sztuba-Solinska, Ph.D.

Academic Editor

PLOS ONE

Journal Requirements:

1. We noticed you have some minor occurrence of overlapping text with your previous publication, which needs to be addressed:

- https://academic.oup.com/nar/article/47/13/7105/5519168

In your revision ensure you rephrase any duplicated text outside the methods section.

2. Thank you for including your competing interests statement; "I have read the journal's policy and the authors of this manuscript have the following competing interests: H.M.A. is an advisor and holds an ownership interest in Nymirum Inc., an RNA- based drug discovery company. "

Reviewers' comments:

Reviewer's Responses to Questions

**Comments to the Author**

1. Is the manuscript technically sound, and do the data support the conclusions?

Reviewer #1: Yes

Reviewer #2: Yes

Reviewer #3: Yes

Reviewer #4: Yes

2. Has the statistical analysis been performed appropriately and rigorously? 

Reviewer #1: I Don't Know

Reviewer #2: N/A

Reviewer #3: I Don't Know

Reviewer #4: Yes

3. Have the authors made all data underlying the findings in their manuscript fully available?

Reviewer #1: Yes

Reviewer #2: Yes

Reviewer #3: Yes

Reviewer #4: Yes

4. Is the manuscript presented in an intelligible fashion and written in standard English?

Reviewer #1: Yes

Reviewer #2: Yes

Reviewer #3: Yes

Reviewer #4: Yes

5. Review Comments to the Author

Reviewer #1: The effect on structure of methylation of A68 in the stemloop IIB of the HIV RRE has been studied in this report using a variety of physical-chemical methods. All the experiments were conducted in vitro with purified RNA substrates. The study clearly shows that the methylation causes no significant change in any of the parameters measured in any of the studies. It also shows that greater effects of methylation can be seen when measurements are conducted in the absence of magnesium, underscoring the importance of carrying out measurements under more physiological conditions. Intriguingly, the authors did show a slight (2 fold) change in the binding of the Rev-ARM to the structure containing the methylation. Contrary to other reports the effect was actually a weakened binding. The authors point out that this is more consistent with what would be expected given the physical effects expected from the methylation. As the authors carefully point out, these cannot studies do not rule out that methylation may effect the full-length RRE struture, full-length Rev protein binding or the binding of other cellular factors differently. This is a weakness of this study but in so far as the authors address this issue clearly and the data presented here is sound, this study seems to fit the criteria needed to be acceptable for PLOS One.

Reviewer #2: The authors present evidence that methylation has minimal effects on the properties of stem IIB of HIV-1 RRE. They use several biophysical techniques in this analysis. This conclusion is interesting and potentially significant.

Reviewer #3: Chia-Chieh Chu and collaborators present a controversial article where they claim that m6A has little impact on the HIV-1 RRE stem IIB structure and in a viral protein binding proprieties. The authors conclude by three biophysical experiments: optical melting experiments, NMR and binding assays that the m6A modification in the position 68 of the the Rev response element is not sufficient to produce significant changes in its conformation. Chu et al, also analyze a functional motif in Rev, one of its best-known proteins that bind RRE and they conclude that m6A does not affect its binding properties. This is a very interesting research article that prevent from dogmas since contradicts some of the literature cited. However, authors need to consider that discarding or dismissing another study requires more evidence. They need to soften some or their conclusions due to the nature of their in vitro studies. They lack description of some important controls for example: in all experiments, how do they make sure the correct “natural” folding of the RRE RNA? How do they know that all the molecules are folded in the same way in vivo? Would it be necessary to consider in vivo crosslinked samples in controls to validate their hypothesis? How do they measure the Rev-RRE association? Finally, they emphasize that Mg+ and the sequence context impact on m6A-RRE RNA structure, but they did not perform any experiments related to the sequence context. More experiments in this matter are necessary to support this conclusion of a less strong sentence needs to be changed to soften this conclusion. In line 256 they establish that “…m6A slightly weakens binding to the Rev-ARM peptide.” Could it be that “slightly” difference in the cell is catalytically important? In my opinion, this paper contains well documented and reliable experiments, it is very well constructed, detailed and has a didactic approach. However strong affirmations i.e. line 286 “This underscores the importance of studying the impact

287 of m6A modifications under conditions that mimic the physiological cellular environment.” Are not congruent with the data presented and need to be state in an ease way.

Reviewer #4: There is some controversy in the field regarding the role of m6A methylation in HIV replication. In particular, the role of m6A methylation sites within the HIV Rev response element (RRE) stem IIB, which have previously been suggested to enhance binding to the HIV REV protein, which in turn enhances REV-mediated viral mRNA nuclear export. Several studies have failed to identify these proposed m6A sites using m6A-seq approaches.

Chu et al., provide convincing evidence using a range of biophyisical approaches, such as UV melting and NMR-based approaches, to show that m6A methylation does not affect the structure of RREIIB. Moreover, flourescence aniosotrophy was used to suggest that m6A in this region actually weakens REV binding.

Specific comments.

1. The introduction could be extended to include other examples of how it can stabilise RNA structures and is implicated in various other RNA processing events.

2. There is no mention of m6A readers.

3. A better explanation of why Mg2+ is brought into the experiments and its significance is required throughout the paper.

4. Line 207 - needs amending

5. The flourescence aniosotrophy experiment is interesting, but very brief, this is a complete opposite to the original finding. Therefore, it needs a confirmatory experiment, as this is just based on a REV-ARM peptide to the RREIIB.

6. PLOS authors have the option to publish the peer review history of their article (what does this mean?). If published, this will include your full peer review and any attached files.

Reviewer #1: No

Reviewer #2: No

Reviewer #3: Yes: Gabriela Toomer

Reviewer #4: No

---

## [Author Response · Author response to Decision Letter 0]

24 Nov 2019

Reviewer #1: 

“The effect on structure of methylation of A68 in the stemloop IIB of the HIV RRE has been studied in this report using a variety of physical-chemical methods. All the experiments were conducted in vitro with purified RNA substrates. The study clearly shows that the methylation causes no significant change in any of the parameters measured in any of the studies. It also shows that greater effects of methylation can be seen when measurements are conducted in the absence of magnesium, underscoring the importance of carrying out measurements under more physiological conditions. Intriguingly, the authors did show a slight (2 fold) change in the binding of the Rev-ARM to the structure containing the methylation. Contrary to other reports the effect was actually a weakened binding. The authors point out that this is more consistent with what would be expected given the physical effects expected from the methylation. As the authors carefully point out, these cannot studies do not rule out that methylation may effect the full-length RRE struture, full-length Rev protein binding or the binding of other cellular factors differently. This is a weakness of this study but in so far as the authors address this issue clearly and the data presented here is sound, this study seems to fit the criteria needed to be acceptable for PLOS One.”

We thank the reviewer for his/her comments.

Reviewer #2: 

The authors present evidence that methylation has minimal effects on the properties of stem IIB of HIV-1 RRE. They use several biophysical techniques in this analysis. This conclusion is interesting and potentially significant.

We thank the reviewer for his/her comments.

Reviewer #3: 

1) “Chia-Chieh Chu and collaborators present a controversial article where they claim that m6A has little impact on the HIV-1 RRE stem IIB structure and in a viral protein binding proprieties. The authors conclude by three biophysical experiments: optical melting experiments, NMR and binding assays that the m6A modification in the position 68 of the the Rev response element is not sufficient to produce significant changes in its conformation. Chu et al, also analyze a functional motif in Rev, one of its best-known proteins that bind RRE and they conclude that m6A does not affect its binding properties. This is a very interesting research article that prevent from dogmas since contradicts some of the literature cited. However, authors need to consider that discarding or dismissing another study requires more evidence. They need to soften some or their conclusions due to the nature of their in vitro studies. They lack description of some important controls for example: in all experiments, how do they make sure the correct “natural” folding of the RRE RNA? How do they know that all the molecules are folded in the same way in vivo? Would it be necessary to consider in vivo crosslinked samples in controls to validate their hypothesis? How do they measure the Rev-RRE association?”

The structure of RRE stem IIB has been the subject of numerous investigations in vitro by X-ray, NMR, SAXS and also more recently in the context of much larger RRE fragments using chemical probing data. In all cases, the structure of RREIIB has been determined to be the same as that observed under our NMR conditions as elaborated in ref 28. We have clarified this point on page 9 on line148 by stating that “Prior X-ray [ref 42, 43], NMR [ref 44, 45], SAXS [ref 46] as well as chemical probing data on larger fragments of the HIV genome (~350 nt) [ref 47] indicate that RREII stem IIB adopts the predominant secondary structure shown in Fig 1A.”

We agree with the reviewer that we cannot rule out that the binding properties may differ in vivo and this is why we noted in the original submission on line 292 that “Our data cannot rule out that the exposed and accessible m6A enhances binding in the context of full-length Rev and RRE or promotes RNA export through the recruitment of other host export factors.” 

We agree with the reviewer that the structural and dynamic properties of RREIIB may differ in vivo. In a recent study [ref 29], we showed that the dynamic properties of RREIIB determined in vitro are very similar to the dynamic properties measured in cells. This argues against a major difference between the two environments at least for unmodified RREIIB. Nevertheless, to soften the language and emphasize this possibility, we added the following sentence to our discussion on page 16 on line 296, “It is also possible that the structural and dynamic properties of m6A modified RREIIB differ in vivo relative to in vitro. A recent study [ref 29] showed that the key dynamic properties of unmodified RREIIB are similar in vitro and in cells. Further studies are needed to examine the behavior of m6A modified RREIIB in cells. “

2) Finally, they emphasize that Mg+ and the sequence context impact on m6A-RRE RNA structure, but they did not perform any experiments related to the sequence context. More experiments in this matter are necessary to support this conclusion of a less strong sentence needs to be changed to soften this conclusion. 

With regards to different sequence contexts, we were referring to other published studies cited in our manuscript [refs 31, 32, 36, 37] which showed that m6A has different effects on RNA stability depending on sequence context and in some cases on the presence/absence of Mg2+. We have clarified this by adding the following sentence at the end of the introduction: “For example, while m6A has been shown to destabilize canonical duplexes [ref 31, 32], it can stabilize junctional A-U base pairs in a Mg2+ and secondary structure dependent manner [ref 37] as well as in contexts in which the m6A is in a dangling end [ref 32].”

3) In line 256 they establish that “…m6A slightly weakens binding to the Rev-ARM peptide.” Could it be that “slightly” difference in the cell is catalytically important? In my opinion, this paper contains well documented and reliable experiments, it is very well constructed, detailed and has a didactic approach. However strong affirmations i.e. line 286 “This underscores the importance of studying the impact

287 of m6A modifications under conditions that mimic the physiological cellular environment.” Are not congruent with the data presented and need to be state in an ease way.

We appreciate the comment made by the reviewer. We agree that we need to be cautious not to over interpret the data measured in vitro as the behavior could change in vivo. In the original submission, we did acknowledge that the in vivo environment could alter the behavior. In the revised manuscript, we made additional changes to soften the language. In the abstract on line 22, we replaced “unlikely to substantially alter the RRE-Rev interaction” with “m6A is unlikely to substantially alter the conformational properties of the RRE.” so that the focus is on the impact of m6A on the conformational properties. 

On line 65, we now note, “The results indicate that if RREIIB is m6A modified, it is unlikely to substantially change the conformational properties of RRE although we cannot rule out that small changes in the conformational properties could modulate Rev-RRE binding in vivo.”

In the revised manuscript, we also edited the following sentence in the discussion section on line 304 “This underscores the importance of studying the impact of m6A modifications in the presence of Mg2+ as in a more physiological conditions.”  

Reviewer #4: 

There is some controversy in the field regarding the role of m6A methylation in HIV replication. In particular, the role of m6A methylation sites within the HIV Rev response element (RRE) stem IIB, which have previously been suggested to enhance binding to the HIV REV protein, which in turn enhances REV-mediated viral mRNA nuclear export. Several studies have failed to identify these proposed m6A sites using m6A-seq approaches.

Chu et al., provide convincing evidence using a range of biophyisical approaches, such as UV melting and NMR-based approaches, to show that m6A methylation does not affect the structure of RREIIB. Moreover, flourescence aniosotrophy was used to suggest that m6A in this region actually weakens REV binding.

Specific comments.

1. The introduction could be extended to include other examples of how it can stabilise RNA structures and is implicated in various other RNA processing events.

We thank the reviewer for this suggestion. We added text to refer to specific examples where m6A can stabilize RNA (see response to comment 2 by Reviewer 3).

2. There is no mention of m6A readers.

In the revised manuscript, we added the following sentence in the introduction on line 44 “Another study employing distinct cell lines and mapping methods did not observe these m6A sites on RREIIB suggesting that m6A can enhance HIV-1 replication and mRNA expression through recruitment of the m6A reader protein YTH-domain containing family (YTHDF) [ref 21]. A third study found m6A in RRE but did not identify the specific site [ref 18]. The study proposed that the YTHDF proteins inhibit HIV-1 replication and infection by blocking viral reverse transcription.” 

3. A better explanation of why Mg2+ is brought into the experiments and its significance is required throughout the paper.

We thank the reviewer for the suggestions. In the revised manuscript, we added the following sentences on line 142 “All experiments were performed in the presence of 3 mM Mg2+ unless stated otherwise. This is important given the impact of Mg2+ on RNA folding and dynamics [ref 41] and also given recent studies showing that the impact of m6A on RNA structural dynamics and stability can depend on Mg2+ [ref 37]. 

4. Line 207 - needs amending

In the revised manuscript, we edited the following sentence in the result section in line 219 (previous line 207) “We recently showed that RREII transiently adopts two low-abundance alternative secondary structures (‘excited states’, ES) referred to as ES1 and ES2 (Fig 3A).” 

5. The flourescence aniosotrophy experiment is interesting, but very brief, this is a complete opposite to the original finding. Therefore, it needs a confirmatory experiment, as this is just based on a REV-ARM peptide to the RREIIB.

The binding affinity for m6A modified was only slightly weakened relative to unmodified RREIIB especially when considering the uncertainty. So our results are best described not as opposite to what was reported previously in ref 20; rather we did not see any evidence for tighter binding with m6A modified RREIIB. We revised the manuscript to soften the language when making this point on page 14 on line 268, “the binding affinity for m6A modified was only slightly weakened relative to unmodified RREIIB especially when considering the uncertainty”. 

In addition, we note that no Kd was measured in the prior study [ref 20]. Rather the tighter affinity was deduced based on ~2-3 fold decrease in pull down of RRE RNA when knocking down two subunits (METTL3/METTL14) from the methyltransferase complex responsible for producing m6A. To address the reviewer comment, we added the following sentence in the discussion on line 292, “Our data cannot rule out that the exposed and accessible m6A enhances binding in the context of full-length Rev and RRE or promotes RNA export through the recruitment of other host export factors, and that this in turn gives rise to the reported ~2-3 fold decrease in HIV viral RNA pull down using Rev when knocking down METTL3/METTL14 responsible for producing m6A [ref 20] ”. 

Editorial

1. We noticed you have some minor occurrence of overlapping text with your previous publication, which needs to be addressed:

- https://academic.oup.com/nar/article/47/13/7105/5519168

In your revision ensure you rephrase any duplicated text outside the methods section.

In the revised manuscript, we edited the text of fluorescence polarization assays in the methods section to avoid overlapping text. 

2. Thank you for including your competing interests statement; "I have read the journal's policy and the authors of this manuscript have the following competing interests: H.M.A. is an advisor and holds an ownership interest in Nymirum Inc., an RNA- based drug discovery company. "

In the revised manuscript, we included “This does not alter our adherence to PLOS ONE policies on sharing data and materials.” in Conflict of interest statement section.

---

## [Editor Report · Decision Letter 1]

27 Nov 2019

m6A minimally impacts the structure, dynamics, and Rev ARM binding properties of HIV-1 RRE stem IIB

PONE-D-19-28951R1

Dear Dr. Al-Hashimi,

We are pleased to inform you that your manuscript has been judged scientifically suitable for publication and will be formally accepted for publication once it complies with all outstanding technical requirements.

With kind regards,

Joanna Sztuba-Solinska, Ph.D.

Academic Editor

PLOS ONE

---

## [Editor Report · Acceptance letter]

4 Dec 2019

PONE-D-19-28951R1 

m^6^A minimally impacts the structure, dynamics, and Rev ARM binding properties of HIV-1 RRE stem IIB 

Dear Dr. Al-Hashimi:

I am pleased to inform you that your manuscript has been deemed suitable for publication in PLOS ONE. Congratulations! Your manuscript is now with our production department. 

With kind regards,

on behalf of

Dr. Joanna Sztuba-Solinska 

Academic Editor

PLOS ONE